**METHODS AND PROTOCOLS**
Novel Systems Biology Techniques

# Quantifying Variation in Bacterial Reproductive Fitness: a High-Throughput Method

Pascal M. Frey,[a,b,c] Julian Baer,[a] Judith Bergada-Pijuan,[a] Conor Lawless,[d] Philipp K. Bühler,[e] Roger D. Kouyos,[a] Katherine P. Lemon,[c,f,g,h] Annelies S. Zinkernagel,[a] Silvio D. Brugger[a,c,i]

[a]Department of Infectious Diseases and Hospital Epidemiology, University Hospital Zurich, University of Zurich, Zurich, Switzerland
[b]Department of General Internal Medicine, Bern University Hospital, University of Bern, Bern, Switzerland
[c]Department of Microbiology, The Forsyth Institute, Cambridge, Massachusetts, USA
[d]Translational and Clinical Research Institute, Medical School, Newcastle University, Newcastle upon Tyne, United Kingdom
[e]Institute for Intensive Care Medicine, University Hospital Zurich, University of Zurich, Zurich, Switzerland
[f]Division of Infectious Diseases, Boston Children's Hospital, Harvard Medical School, Boston, Massachusetts, USA
[g]Alkek Center for Metagenomics & Microbiome Research, Department of Molecular Virology & Microbiology, Baylor College of Medicine, Houston, Texas, USA
[h]Section of Infectious Diseases, Department of Pediatrics, Texas Children's Hospital and Baylor College of Medicine, Houston, Texas, USA
[i]Department of Oral Medicine, Infection, and Immunity, Harvard School of Dental Medicine, Boston, Massachusetts, USA

Pascal M. Frey, Julian Baer, and Judith Bergada-Pijuan contributed equally; the order was determined by the sequence of joining the project.

**ABSTRACT**  To evaluate changes in reproductive fitness of bacteria, e.g., after acquisition of antimicrobial resistance, a low-cost high-throughput method to analyze bacterial growth on agar is desirable for broad usability. In our bacterial quantitative fitness analysis (BaQFA), arrayed cultures are spotted on agar and photographed sequentially while growing. These time-lapse images are analyzed using a purpose-built open-source software to derive normalized image intensity (NI) values for each culture spot. Subsequently, a Gompertz growth model is fitted to NI values, and fitness is calculated from model parameters. To represent a range of clinically important pathogenic bacteria, we used different strains of *Enterococcus faecium*, *Escherichia coli*, and *Staphylococcus aureus*, with and without antimicrobial resistance. Relative competitive fitness (RCF) was defined as the mean fitness ratio of two strains growing competitively on one plate. BaQFA permitted the accurate construction of growth curves from bacteria grown on semisolid agar plates and fitting of Gompertz models. Normalized image intensity values showed a strong association with the total CFU/ml count per spotted culture ($P < 0.001$) for all strains of the three species. BaQFA showed relevant reproductive fitness differences between individual strains, suggesting substantially higher fitness of methicillin-resistant *S. aureus* JE2 than Cowan (RCF, 1.58; $P < 0.001$). Similarly, the vancomycin-resistant *E. faecium* ST172b showed higher competitive fitness than susceptible *E. faecium* ST172 (RCF, 1.59; $P < 0.001$). Our BaQFA method allows detection of fitness differences between bacterial strains and may help to estimate epidemiological antimicrobial persistence or contribute to the prediction of clinical outcomes in severe infections.

**IMPORTANCE** Reproductive fitness of bacteria is a major factor in the evolution and persistence of antimicrobial resistance and may play an important role in severe infections. With a computational approach to quantify fitness in bacteria growing competitively on agar plates, our high-throughput method has been designed to obtain additional phenotypic data for antimicrobial resistance analysis at a low cost. Furthermore, our bacterial quantitative fitness analysis (BaQFA) enables the investigation of a link between bacterial fitness and clinical outcomes in severe invasive bacterial infections. This may allow future use of our method for patient management and risk stratification of clinical outcomes. Our proposed method uses open-source software and a hardware setup that can utilize consumer electronics. This will enable

Address correspondence to Pascal M. Frey, pascal.frey@insel.ch, or Silvio D. Brugger, silvio.brugger@usz.ch.

a wider community of researchers, including those from low-resource countries, where the burden of antimicrobial resistance is highest, to obtain valuable information about emerging bacterial strains.

**KEYWORDS** quantitative fitness analysis, antimicrobial resistance, bacterial fitness, image analysis

Reproductive fitness, i.e., the capacity to reproduce in ideal circumstances, is an important predictor of the evolutionary success of a bacterial genotype. Therefore, quantifying changes in fitness is highly informative about the evolutionary potential of bacterial genotypes.

Antimicrobial-resistant strains of pathogenic bacteria with the ability to cause severe infections in humans are widely regarded as a threat to public health. Although research is often focused on the development and acquisition of antimicrobial resistance, it is generally thought that most acquired antimicrobial resistances come with a cost in reproductive fitness, and thus they are expected to disappear if selection pressure from antibiotics is reduced (1). However, epidemiologic evidence to support this assumption is scarce, and the routine use of methods to screen for changes in the reproductive fitness of emerging antimicrobial-resistant bacterial strains is not widely applied, possibly due to a lack of availability of high-throughput and low-cost methods, especially in low-resource settings that show a high prevalence of antibiotic resistance (2).

Furthermore, a high-throughput method to accurately determine reproductive fitness in bacteria despite an expectedly high in-strain phenotypic variability may not only be an important instrument in predicting the epidemiologic persistence of resistant strains (1), but might also provide a useful insight regarding clinical outcomes of severe infections in general. In order to investigate the fitness of bacteria in smaller laboratories and in settings with limited resources, a low-cost high-throughput method is needed to cope with the large amount of data required for routine analysis.

A similar requirement has previously driven the design of automated synthetic genetic array (SGA) methods, where interactions of genes, usually in yeast as a model eukaryotic species, are studied on a large scale while allowing interaction between mutants during growth (3). Among other approaches, this need has also led to the development of an automated high-throughput quantitative fitness analysis (QFA) method for yeast, which was used as a eukaryotic model for the investigation of reproductive fitness of human cell mutations associated with telomere capping (4). In QFA, an array of yeast cultures (e.g., 8 by 12, or 16 by 24) with different mutants is spotted onto a rectangular agar plate in a predefined pattern, where every single culture spot is then growing in competition with its neighboring mutant. During growth, time-lapse photographs of the agar plate with its cultures are taken in a defined time interval. These images are then processed using purpose-built open-source software, which derives intensity values for each culture spot (5) as a surrogate measure for each spot's microbial population density. These intensity values are subsequently analyzed using a designated R package (6), which fits a logistic growth model over measured time-lapse values and derives fitness from parametrization of the mathematical model (4, 5).

A similar method for quantifying the fitness of *Escherichia coli* mutants, "colony-live," has also been developed (7). This method is well validated and accurate; however, it relies on one dedicated scanner per plate, with a scanning illumination that is placed to measure the absorbance of growing colonies in a way that is unsuitable for measuring the reflection of light from the colonies as required with regular photo cameras. The dependence of the colony-live method on expensive specialized equipment leaves a need for a low-cost and flexible method to be automated with consumer electronics.

However, the principles of the QFA method are considered to be applicable to bacteria as well (4). The QFA setup is flexible, and the time-lapse image analysis algorithm is designed to read normal photographs from a consumer electronics digital single-lens reflex (DSLR) camera. Thus, we aimed to adapt the QFA method for use with *Staphylococcus aureus* and other important bacterial pathogens in human medicine, using low-cost consumer products for basic QFA automation and data acquisition. The development of a bacterial QFA (BaQFA) method and the validation of its use with bacteria fills the gap of an inexpensive, versatile, open-source, and easy-to-use option designed to analyze bacterial fitness (Table 1).

We hypothesized that based on the QFA approach for yeast, we could develop a new method for bacteria that would yield valid bacterial reproductive fitness results accurate enough to detect differences between strains with different antimicrobial resistance properties despite an expectedly high random biological variation.

## RESULTS

**Experimental design and workflow.** For our bacterial quantitative fitness analysis (BaQFA) method, standardized inocula of different bacteria were suspended in phosphate-buffered saline (PBS) solution and transferred onto brain heart infusion (BHI) agar using a 96-well plate and a 96-pin replica plater. After inoculation, the BHI agar plate was incubated at 37°C, and automated time-lapse photos of the 96 culture spots were taken during growth. From these photos, we derived culture intensity for each spot over time, fitted a Gompertz growth model to the data, and derived fitness estimates from the model parameters (Fig. 1).

Overall, we analyzed the fitness from 2,160 single culture spots for the development of our BaQFA method. After the adjustments to the culture spotting routine and the computational processing using the new BaColonyzer software as well as BaQFA with Gompertz model fitting, we were able to apply the modified BaQFA method to all *S. aureus* and *Enterococcus faecium* strains. For *Escherichia coli*, we successfully derived intensity values and their correlation to CFU counts, ensuring full functionality with this Gram-negative bacterium.

**Derived normalized intensity values and CFU counts.** In order to obtain the culture growth intensity measurements, we developed an open-source software named BaColonyzer to process, normalize, and analyze time-lapse images (Fig. 2). When validating the BaColonyzer measurements, we found a strong association between CFU counts from single culture spots and the respective normalized intensity measurements from BaColonyzer taken just before determination of CFU counts. Associations for all examined strains were approximately log-linear (Fig. 3). Bland-Altman plots comparing measurements of the same bacterial cultures with Colonyzer software for yeast and the new BaColonyzer suggested that BaColonyzer measures the normalized intensity on a larger range, with smaller values in low-intensity cultures and larger values in high-intensity cultures (Fig. S6).

**Reproductive fitness with and without antibiotic resistance. (i) *S. aureus* strains Cowan and JE2.** As proof of principle that we could detect fitness differences of strains with various antibiotic resistance properties, we compared two strains of *S. aureus*. Different fitness, defined as higher maximal growth rate (MGR) and lower time to MGR ($T_{max}$), was observed for the tested strains of *S. aureus*, with JE2 showing higher fitness (Table 2) and a broader biologic variability of fitness than Cowan (Fig. 4).

When grown competitively in the absence of antibiotics, we found strong evidence for a lower fitness of the methicillin-susceptible Cowan strain compared to the methicillin-resistant JE2 strain (overall relative competitive fitness [RCF], 1.58 confidence interval [CI], 1.51 to 1.66; $P < 0.001$). Between three replications of BaQFA, the RCF varied from 1.56 (CI, 1.43 to 1.69) to 1.71 (CI, 1.61 to 1.82) to 1.48 (CI, 1.37 to 1.60) for replicates 1, 2, and 3, respectively. The presence of the Cowan strain on a grid layout increased fitness for JE2, while JE2 decreased the fitness of Cowan, a sign of possible interaction (Fig. 4). An additional comparison of competitive fitness of the Cowan and JE2 strains with the clinical isolate is given in Fig. S2.

**TABLE 1** Comparison of different software to analyze growth of multiple microbial cultures growing on an agar plate

| Software | Required hardware cost[a] | End-to-end | Grid flexibility | Corrections | Growth fit | Availability | Installation | Validated for bacteria |
|---|---|---|---|---|---|---|---|---|
| BaQFA | Inexpensive | Yes | High: any grid | Image stabilization, lighting correction, reference normalization, image renaming | Multiple built-in models Default: Gompertz model | Open source | Very easy | Yes |
| QFA | Inexpensive | Yes | High: any grid | Lighting correction | Logistic model | Open source | Difficult | No |
| Colony-live | Inexpensive | Yes | Unclear: 32 × 48 grid | Neighboring correction, maximum growth rate correction | Gompertz model | Upon request | Not tested | Yes |
| YeastXtract | Inexpensive | Yes | Rigid: 8 × 12 grid | Image stabilization, spot area normalization | Logistic model | Upon request | Not tested | No |
| Pyphe | Inexpensive | Yes | Low: few grid possibilities | Spatial correction (including grid normalization and row/column median normalization), correction of artifacts | 2 options: maximum slope, endpoint colony size | Open source | Easy, but permission issues arise | No |
| CellProfiler | NA | No | 8 × 12 grid | Unclear | None | Open source | Very easy | Unclear |
| gitter | NA | No | High: any grid | Global lighting, small rotations, identification threshold | None | Open source | Not tested | Partially |

[a]NA, not available (no dedicated hardware setup).

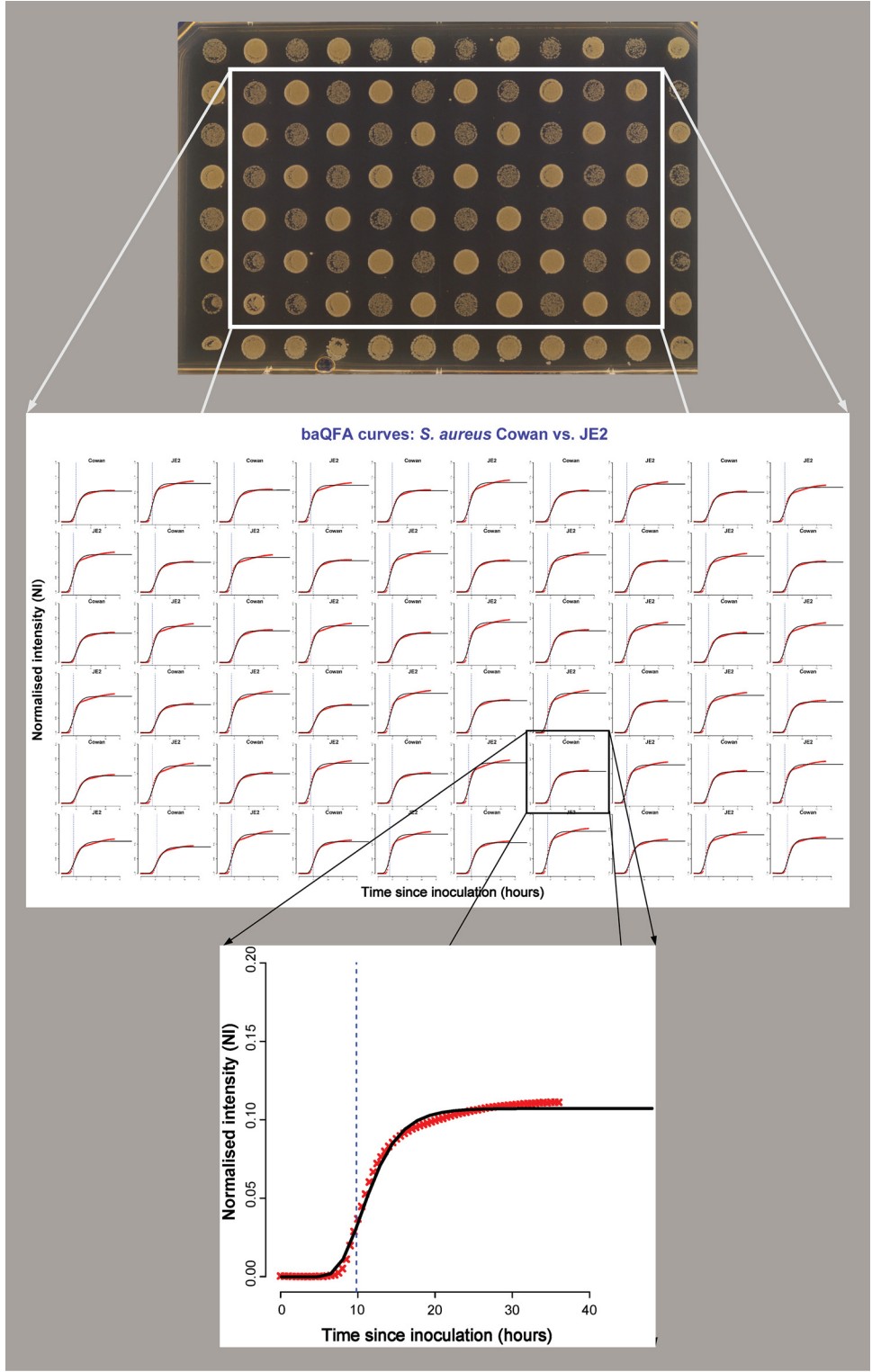

**FIG 1** Construction of growth curves and fitness measures from time-lapse photos of *S. aureus* strains Cowan and JE2. After discarding the culture spots at the plate borders, the intensity measures for each time point are derived from time-lapse images using the custom-built BaColonyzer software. Using the BaQFA package for R, growth curves and fitness measures can then be calculated from the normalized intensity measures. The blue dotted vertical line marks the point of maximum growth rate.

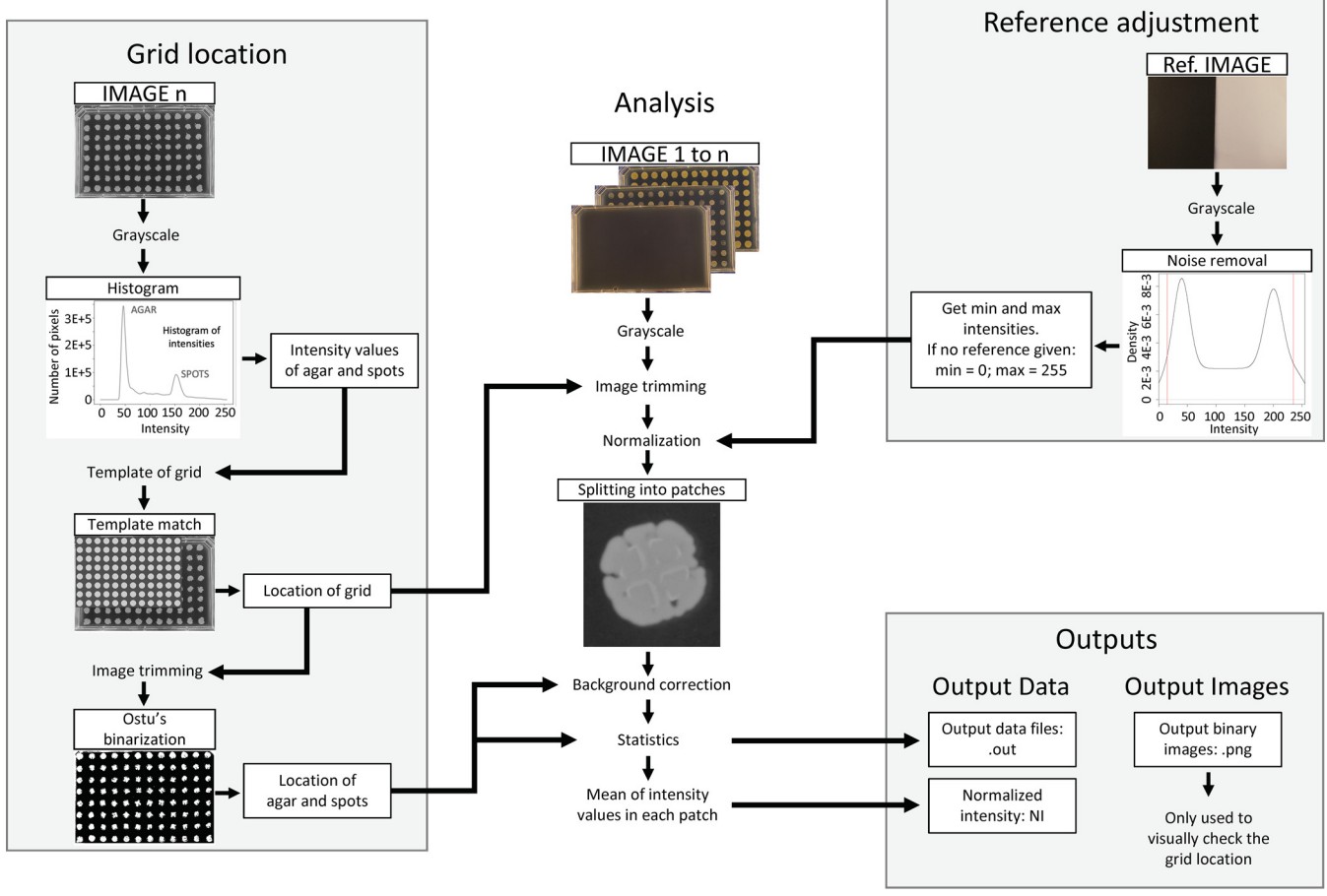

**FIG 2** Conceptual design of the BaColonyzer algorithm, described in more detail in Text S1.

**(ii) *E. faecium* strains ST172 and ST172b.** Noncompetitional growth of ST172 and ST172b, where all culture spots on a plate consisted of only one strain (solo), did not reveal substantial differences in fitness between the plates, with considerable between-plate variance. However, when grown in direct competition, with culture spots of each strain arrayed in a grid next to each other, we observed a higher fitness of the vancomycin-resistant *E. faecium* strain ST172b than that of vancomycin-susceptible strain ST172 (Fig. 5), where the ST172b was, on average, 1.59 times more fit than ST172 (Table 2). *E. faecium* ST172 and ST172b are closely related clinical isolates. The core genome of both strains is identical (no variants detected), but the two differ in their accessory genome. ST172b contains the genes of the *vanA* operon and 12 other predicted genes that are not present in isolate ST172. On the other hand, ST172 contains 32 predicted genes that ST172b does not have. Those strains were chosen to compare relative fitness in closely related strains differing phenotypically in antibiotic resistance.

**(iii) Growth models and fitness measures.** Compared to a standard or generalized logistic growth model, the Gompertz model showed an overall good model fit (Fig. S1A), while also featuring the lowest sum of squared error of the three models (Fig. S1B). Testing many different parameters of growth and fitness, we did not observe substantially different qualitative results, with the *S. aureus* JE2 and *E. faecium* ST172b strains showing consistently higher fitness than Cowan or ST172 strains, respectively, when grown in competition (Fig. S3).

When *S. aureus* strains JE2 and Cowan were grown together in liquid BHI, JE2 was more abundant than Cowan after 24 h (JE2 proportion at t = 0 h: mean ± standard error 67.1 ± 5.85%, t = 24 h: 91.9 ± 2.87%, $P = 0.034$, $n = 3$), as predicted by our BaQFA results (Fig. S4).

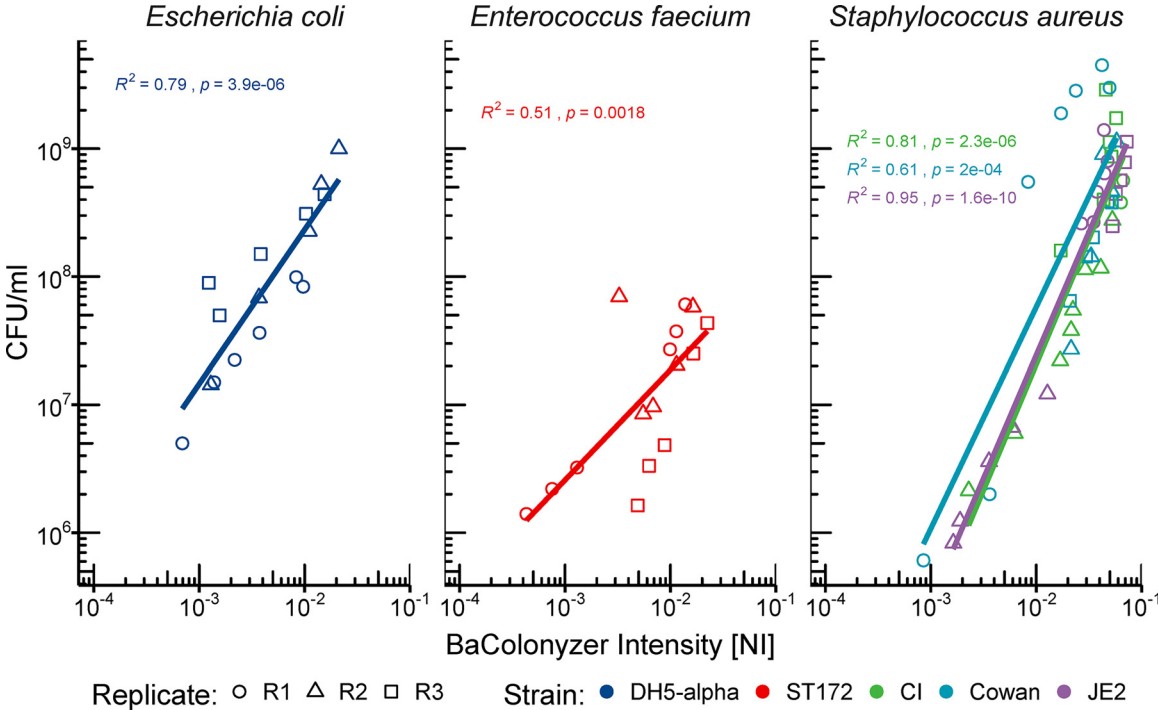

**FIG 3** Correlation of CFU and BaColonyzer-derived intensity measures. The relationship of BaColonyzer normalized intensity (NI) measurements and CFU per milliliter (CFU/ml) counts was approximately log-linear for each strain and showed some variation between the different strains. The linear regression found a strong association of NI and CFU counts ($P < 0.01$). Each dot represents the mean of three technical replicates from the same time point.

## DISCUSSION

Inspired by a previously established technique of quantitative fitness analysis (QFA), we developed and validated our BaQFA method to quantify fitness of bacterial strains. Based on our own interests, we used important pathogenic bacterial strains with and without antimicrobial resistance. However, our BaQFA method is broadly applicable for quantifying the fitness of bacterial strains from any habitat of interest.

We were able to capture meaningful results from our BaQFA method and detected differences in reproductive fitness, defined as the capacity to reproduce in ideal circumstances, between certain strains with and without antimicrobial resistance. We did not intend to answer the question of whether the antimicrobial resistance alone was causal for the observed differences in fitness and, thus, did not further examine isogenic strains with and without antimicrobial resistance mutations. Our newly developed BaColonyzer software showed a slightly larger scale range of normalized intensity (NI) values compared to the previous Colonyzer software, illustrated by a linearly decreasing slope in the Bland-Altman plots. Although we did not intend for this specifically during development of the new algorithm, we considered the larger scale to potentially improve detection of smaller differences in fitness between strains. Compared to a standard logistic model or generalized logistic growth model, the Gompertz model was the most

**TABLE 2** Absolute and relative competitive fitness when growing in competition[a]

| Species | Strain | Mean fitness[b] (SD) | RCF (95% CI) | P value |
|---|---|---|---|---|
| Staphylococcus aureus | Cowan | 1.3142 (0.2232) | Reference | |
| Staphylococcus aureus | JE2 | 2.0785 (0.3472) | 1.58 (1.51–1.66) | <0.0001 |
| Enterococcus faecium | ST172 | 0.3802 (0.1148) | Reference | |
| Enterococcus faecium | ST172b | 0.6054 (0.1884) | 1.59 (1.45–1.74) | <0.0001 |

[a]SD, standard deviation; RCF, relative competitive fitness; CI, confidence interval; NI, normalized intensity.
[b]Given as mNI/h².

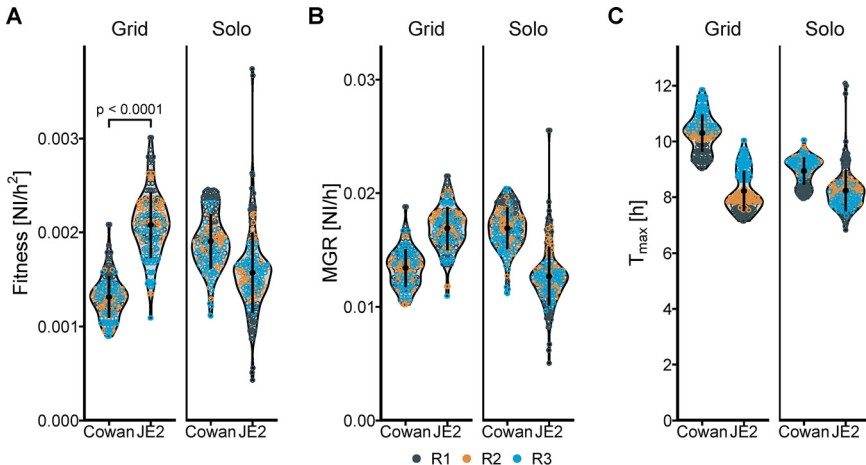

**FIG 4** Fitness of methicillin-resistant *S. aureus* (MRSA) strain JE2 compared to the methicillin-susceptible strain Cowan when grown on agar either alone or together in a grid pattern. *S. aureus* strains JE2 and Cowan grown on agar either alone (Solo) or together in a grid where each JE2 has only Cowan as direct neighbors during growth. (A to C) Fitness (A) is expressed as the maximum growth rate (B) divided by the time to maximum growth (C). Each dot represents one bacterial colony, and the color of the dots represents the different replicates. NI, normalized intensity; MGR, maximum growth rate; $T_{max}$, time to maximum growth rate.

adequate, especially for *S. aureus*. Using different alternative mathematical fitness definitions did not change the qualitative results.

Interestingly, we could observe a potentially important effect of interaction when *S. aureus* strains JE2 and Cowan were grown competitively on one plate, with JE2 consistently reducing the fitness of Cowan while showing increased fitness during this competition. However, determining whether this was due to outcompeting the neighboring strain for nutrients, or if there was another metabolic interaction between the strains, was not an objective of our BaQFA experiments. It could still be hypothesized that with our BaQFA method, bacterial interactions might be quantified, as they have been described similarly *in vivo* and *in vitro* (8, 9). Although we regard the results from our method validation work to only be of an exploratory nature, we could observe a

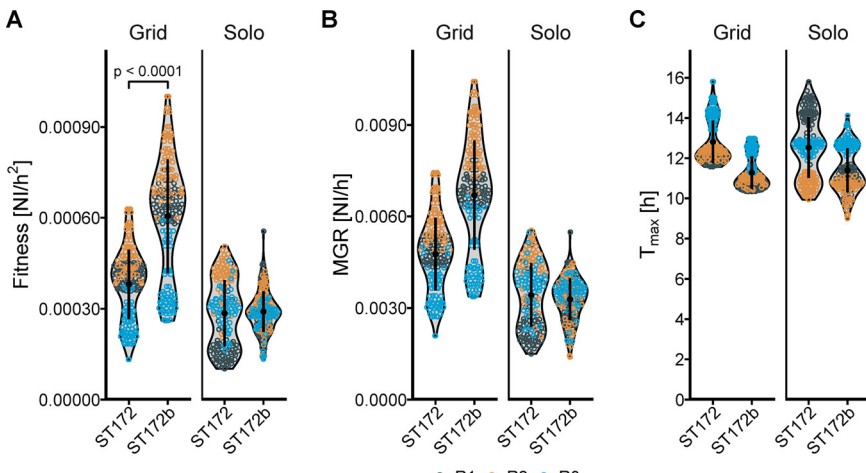

**FIG 5** Fitness of vancomycin-sensitive *E. faecium* strain ST172 compared to the vancomycin-resistant (VRE) strain ST172b when grown on agar either alone or in a grid pattern. Similar to *S. aureus*, BaQFA showed that antibiotic resistance in *E. faecium* ST172b was associated with a gain in fitness compared to the closely related vancomycin-susceptible ST172. (A to C) Fitness (A) is expressed as the maximum growth rate (B) divided by the time to maximum growth (C). Each dot represents one bacterial colony, and the color of the dots represents the different replicates. NI, normalized intensity; MGR, maximum growth rate; $T_{max}$, time to maximum growth rate.

substantially increased fitness in the methicillin-resistant JE2 strain compared to the methicillin-susceptible Cowan strain, suggesting a different baseline-fitness of the underlying USA300 background in JE2 compared to Cowan or the presence of compensatory mutations or other mechanisms of fitness gain. Interestingly, the *E. faecium* strain ST172b, which differs from ST172 mainly by the presence of a vancomycin-resistance plasmid, was more fit in competitive BaQFA, hinting at a possible fitness gain with the vancomycin-resistance plasmid or compensatory mutations. Although acquisition of antibiotic resistance is generally linked to fitness cost and subsequent lower fitness under missing selection pressure (i.e., antibiotic exposure), fitness costs might be alleviated by compensatory changes within the bacteria if they are maintained for generations (10, 11). In the case of vancomycin-resistant enterococci (VRE), however, the persistence of glycopeptide resistance is well described despite missing selection pressure (12, 13). Resistance-containing plasmids can ensure their own maintenance in the absence of antibiotic resistance (12). It has also been speculated that insertion sequence element insertions in the *vanA* gene cluster can result in fitness gain in the absence of glycopeptide exposure (11). As the difference between the closely related ST172 and ST172b is not only the *vanA* cluster, but also several other predicted genes, it could be hypothesized that some of those other differences may provide an additional fitness gain or competitive advantage (e.g., bacteriocins).

Thus, another very powerful application of our BaQFA method is its combination with genomic data analysis. BaQFA may be used to investigate changes of bacterial fitness of certain genomic variants or the interaction of several genomic changes to determine compensatory mutations after antibiotic resistance acquisition.

Regarding cost, the BaQFA-specific materials can be obtained for less than $1,000, provided the lab is equipped with materials for bacterial growth (including an incubator, growth medium, a replica plater, and agar plates); if the consumer electronics are bought second hand, the cost can be lowered to less than $500.

Our BaQFA method has several strengths. First, it allows for an accurate determination of reproductive fitness with high throughput and at a very low cost, while being based on an established and validated technique. Second, the possibility of growing two strains in a direct neighborhood permits competitive effects to be investigated. Third, analyzed bacteria are grown on semisolid medium, allowing for colony behavior such as quorum sensing as a possible factor in reproductive fitness. Finally, we have successfully used low-cost consumer electronics to automate the image capturing and data generation processes, making the method attractive for smaller labs, as well as research groups with a low budget for exploratory research, and investigators from low-resource settings.

Furthermore, our newly designed open-source software and R packages BaColonyzer and BaQFA are easy and fast to install. They also feature much faster analyses; BaColonyzer takes less than 1 s per image on an average laptop, about a fifth of the time the Colonyzer software needed. Additionally, users can benefit from the software to correct and stabilize images in time-series experiments, making the results less sensitive to accidental movements or rotations. To also increase usability, the software can now automatically rename the photo image files to a standard format using the camera image metadata.

The BaQFA method also has several limitations. First, the BaColonyzer software, like its predecessor, is not yet able to correct for background color changes during the time of growth, possibly making it difficult to accurately measure intensity for species such as *Pseudomonas*, which produce colored metabolites. Second, a direct measurement of bacterial species with a low optical signal (e.g., *Streptococcus* spp.) may not produce high enough visibility to be adequately detected on photographic images, while the use of blood agar for enhanced contrast is not yet possible due to the change in color over time. Third, we did not intend to compare different species and mainly focused on *Staphylococcus aureus* strains for data generation, thus deeming the resulting knowledge about the tested bacteria to be only of an exploratory nature. Fourth, we observed some biologic growth variability between culture spots on the same plate, the extent of which differed by species, possibly requiring a larger number

of culture spots per strain in species with high variability and a small difference in fitness between strains. Finally, the absolute fitness measurements showed a rather high between-plate variation, which makes it difficult to universally standardize over different assays and laboratories. Although we could not discern the exact origin of the observed variation, it is most likely caused by slight differences in starting conditions on different plates. Even with our extent of standardizing the BaQFA experiments, it is almost impossible to get the exact same inoculum for different plates, while the plate itself may vary slightly in nutrition concentration and volume. Inherently, the results of growth measurements may change more pronounced after exponential growth, depending on even small differences in starting conditions.

However, the main focus of our method is to detect differences in fitness when bacteria are grown in competition with each other, which is only possible when they are grown on the same plate and is not affected by the between-plate variability. We expect this relative competitive fitness to be less influenced by systematic factors, as they will affect both strains on the plate similarly, possibly making this relative measure more generalizable.

Overall, our low-cost and high-throughput BaQFA method might be an important instrument for the prediction of epidemiologic persistence in the emergence of new resistant bacterial strains, as fitness has been proposed to be a major factor in the evolution of antimicrobial resistance (14).

For direct clinical application, a fast and inexpensive determination of fitness could be a useful cofactor to predict clinical outcomes in complex serious infections or help clinicians in their choice and duration of antimicrobial therapies, adding the reproductive fitness of the pathogen to the various host and pathogen factors being considered for optimal antimicrobial therapy concepts. Furthermore, BaQFA might be used to evaluate the antibacterial activity of bacteriophages, as well as measuring phenomena such as bacterial in-host evolution with loss of fitness after resistance acquisition and regain of fitness through compensatory mutations.

**Conclusion.** In conclusion, we have developed a quantitative fitness analysis method to be used with important pathogenic bacteria. We found that this method provides valid fitness measures. The flexibility to automate the data generation with consumer electronics makes our method accessible for low-resource settings. This paves the way for further research focused on examining associations between reproductive fitness of bacterial pathogens and clinical outcomes or epidemiological persistence of problematic bacterial strains, as reproductive fitness may be a major factor in these complex interactions (14).

## MATERIALS AND METHODS

**Quantitative fitness analysis (QFA) and bacterial pathogens.** Our bacterial QFA (BaQFA) method consisted of cultures of the microbe of interest to be spotted onto solid agar medium in a rectangular array of small neighboring individual culture inoculations. The growth of these neighboring inocula was then followed over time by automated high-resolution time-lapse photography. The images were processed using our custom-built open-source software, BaColonyzer, to derive normalized intensity values. A mathematical model of population growth was subsequently fitted to these data in order to estimate growth parameters, from which reproductive fitness was derived. In our BaQFA method for bacteria, we used a Gompertz growth model for derivation of reproductive fitness parameters (Fig. 1).

To represent a range of important bacterial pathogens in humans, we assessed the accuracy of BaQFA using mainly *Staphylococcus aureus* (methicillin-resistant JE2; methicillin-susceptible Cowan), but also *Enterococcus faecium* (ST172, ST172b) and *Escherichia coli* DH5$\alpha$ (Table 3).

**Microbial culture array and growth conditions.** In preparation for BaQFA, the bacterial strains of interest were streaked out on Columbia sheep blood agar (CSBA) plates and incubated at 37°C for 18 to 24 h. Freshly grown strains were then harvested from the plate with sterile loops and resuspended in phosphate-buffered saline (PBS). After vortexing, the solution was diluted to an optical density (OD$_{600nm}$) of 0.1 ($\pm$0.01).

To prepare the final dilution, 1 ml of this 0.1 OD$_{600}$ solution was diluted in 40 ml PBS (or for lower quantities, 0.5 ml in 20 ml to receive the same final concentration). We then added 200 $\mu$l of this final fresh bacterial suspension to each well of a 96-well plate with either one strain in all wells for solo BaQFA or in any other desired pattern (e.g., a checkerboard grid; placing each tested strain in direct neighborhood of a competing second strain tested for a 2-strain array).

Using a 96-pin replica plater, bacteria were finally transferred from the 96-well plate to a rectangular

**TABLE 3** Overview of bacterial strains used for BaQFA with their antimicrobial resistance

| Species | Strain | Antimicrobial resistance |
|---|---|---|
| Staphylococcus aureus | Cowan | |
| Staphylococcus aureus | JE2 | Methicillin (MRSA) |
| Staphylococcus aureus | Clinical isolate | |
| Enterococcus faecium | ST172 | |
| Enterococcus faecium | ST172b | Vancomycin (VRE) |
| Escherichia coli | DH5$\alpha$ | |

single-well BHI agar plate (20 ml agar medium, Thermo Scientific OmniTray single well with lid). The BHI agar plate was then put into the BaQFA imaging setup (Fig. S5) and incubated at 37°C ambient air (no additional $CO_2$). No antibiotics were added to any of the plates.

While the solo (noncompetitional) experimental design gives an overview of a single strain's undisturbed growth and may be used mainly to observe within-strain variance of fitness, the differences of fitness between strains is investigated by growing them in direct competition with each other on the same plate in a grid pattern, where each culture spot is placed in the direct neighborhood of the competing strain.

**Culture image capturing.** Image capturing was automated using a LEGO Mindstorms EV3 robot, assembled into a purpose-built PVC box within a standard incubator. For plate illumination, we used a consumer electronic cold white LED stripe.

Mounted on a Manfrotto tabletop tripod (kit 209+492), we used a Canon EOS D650 DSLR camera with a Canon EF 40-mm f/2.8 STM lens for imaging. Images were taken at a resolution of 5184 by 3456 pixels, with an ISO-400, aperture f/22 and exposure of 2.5 s in manual mode. The camera shutter was controlled through the LEGO Mindstorms robot and programmed to release every 10, 20, or 30 min after the robot had opened the plate lid (to reduce reflection and condensation artifacts; the lid is needed to keep agar from drying). A picture of the BaQFA experimental setup is given in Fig. S5. The LEGO Mindstorms program is open-source and available on GitHub (github.com/BaQFA).

**Image segmentation and normalized intensity values of bacterial growth.** For the derivation of normalized intensity (NI) values of each culture spot on the time-lapse image series, the open-source software BaColonyzer was created using Python3.

BaColonyzer has three main commands available for the users. The first command, `bacolonyzer rename_images`, allows to rename the image files according to a recommended date-time format (QFAxxxxxxxxxx_YYYY-MM-DD_hh-mm-ss.jpeg, where xxxxxxxxxx is a unique plate ID based on the capture time of the first image). The second command, `bacolonyzer stabilize_images`, allows to stabilize sets of images that may accidentally have been moved or rotated during the experiment, ensuring the greatest precision and reliability of the results. The third command, `bacolonyzer analyze`, is the main command and is needed to analyze time-series of QFA images.

Inspired by Colonyzer (5), the new algorithm is much faster and more robust to handle bacterial colonies. When using the main command, BaColonyzer imports a grayscale version of the images using OpenCV (OpenCV, 2015; Open Source Computer Vision Library), and the final NI is based on the intensity values of the pixels, also accounting for colony size. Specifically, BaColonyzer finds the location of the plate by creating an artificial template of the grid, which is then matched to the last image in the time-lapse series. This allows removal of the borders and other pixels that might introduce noise or artifacts. Otsu's binarization, followed by a dilation of the white pixels, is used to identify the culture spot areas and the agar areas. For a proper NI quantification, BaColonyzer divides the agar plate into smaller patches that contain one colony spot. Next, it normalizes the intensity values of each patch by subtracting the mean intensity of the agar in this patch, thus correcting for color differences within and between images. The NI of each colony is finally computed as the sum of all intensity values of the patch, divided by the number of pixels. Since all patches have exactly the same number of pixels, bigger colonies will result in higher normalized intensities, allowing the user to account also for growth in colony size. Furthermore, BaColonyzer includes an optional normalization that accounts for the camera settings, improving the comparison of various sets of images. A visualization of the conceptual design is given in Fig. 2, while the algorithm is described in more detail in Text S1 and Table S1. BaColonyzer is available as a package from PyPI for Python3, which ensures a fast and easy installation. Documentation, instructions and the source code can be found online on the main documentation webpage as well as on GitHub (judithbergada.github.io/bacolonyzer and github.com/baQFA).

To investigate the assumption of our derived intensity values as a surrogate for the true bacterial population density in a culture spot, we examined the association of computationally derived NI measures and bacterial CFU counts of the same culture spot for three bacterial species. For this, an agar biopsy specimen of the culture spot was vortexed in 1 ml PBS to suspend the bacteria. From this suspension, we performed a dilution series on CSBA plates to estimate the total number of bacterial cells (per ml) on a culture spot. These CFU counts were then matched to the BaColonyzer NI value of the same spot taken before biopsy.

After logarithmic transformation of both the NI values as well as the CFU counts, we used Pearson correlation to determine the association between NI values and CFU counts.

To compare the performance of the Colonyzer software for yeast to our newly built BaColonyzer for image segmentation of bacterial growth, we ran the same culture spot images through each software and compared intensity values for the same spots with the Bland-Altman method (15).

**Growth model parametrization and fitness derivation.** In order to estimate fitness, we used a variation of the Gompertz growth model (16), which was fitted to the normalized intensity values over all time points for each culture spot. Fitness measures were then derived from parametrization of the model. Our variation of the Gompertz model can be defined by a carrying capacity parameter K, absolute maximum growth rate MGR, and the time $t$ to MGR ($T_{max}$), where the population size is $W(t)$ (16):

$$W(t) = K * exp\left(-exp\left(-\frac{e* MGR* (t - T_{max})}{K}\right)\right) \tag{1}$$

Because of the asymmetrical shape of the bacterial growth curves, the fitting of a logistic model—like in the yeast QFA method with yeast—is imperfect. We decided to switch to a Gompertz model, which is well established for bacterial growth (7, 16).

Following Addinall et al. (4), we calculated fitness from two measures of growth: a more simplistic nonlogarithmic maximum absolute growth rate (MGR) and the time to the maximum absolute growth rate ($T_{max}$), where fitness was defined as MGR divided by $T_{max}$ (Equation 2). With these adaptations, a strain with a larger growth rate and a shorter time to maximum growth would reflect a higher reproductive fitness. Similar fitness estimates have been established previously (1, 4, 7, 17).

$$Fitness = \frac{MGR}{T_{max}} \tag{2}$$

The previous yeast QFA method included an R package (qfaR) for model fitting and calculation of fitness measures. This R package has been renamed BaQFA and adapted to fit the need of using it with bacteria, including the new Gompertz model and fitness measure described above. BaQFA is open-source and available on GitHub (github.com/BaQFA).

The BaQFA R package still allows a user to select standard and generalized logistic models, the only options in the yeast qfaR package, to be fitted to the data. Additionally, we added options to specify upper and lower boundaries for all model parameters and the desired time format in hours or days. The user is also not bound to utilize our default fitness definition and can combine various model parameters to a singular fitness parameter reflecting best the given biological question. A selection of possible fitness options is shown in Fig. S3.

**Fitness comparisons and statistical analyses of fitness.** Derived fitness measures from BaQFA were aggregated as mean (SD) fitness, and a density function to represent the fitness distribution per tested strain was visualized using extended violin plots. Since the application of fitness is relative (1), we introduced a relative competitive fitness measure to better quantify differences between two strains when grown in a grid pattern of direct competition to each other on one plate. For this, the ratio of mean fitness was used, with Fieller's method to calculate exact 95% confidence intervals (18).

In a conceptual sensitivity analysis, we compared our Gompertz model to standard and generalized logistic models and used different alternative fitness measures for comparison. Furthermore, we grew the two *S. aureus* JE2 and Cowan strains together in 5 ml liquid BHI medium over 24 h (37°C, 220 rpm), utilizing an adapted inoculation protocol for BaQFA resulting in the same inoculation ratio, to see whether the outcome of this direct competition in liquid was compatible with our estimated competitive fitness from BaQFA. The mixed cultures were serially diluted at $t = 0$ h and $t = 24$ h and spread-plated on CSBA plates. The proportion of JE2 was determined using ColTapp software, with which total CFU were counted and JE2 colonies were identified by their intense hemolysis (19).

All results were considered statistically significant at an alpha level of 0.05. All fitness analyses were performed using R 3.6 (R Core Team; R Foundation for Statistical Computing; 2019).

**Data availability.** The time-lapse picture data sets used and analyzed for our study are available on Figshare (https://doi.org/10.6084/m9.figshare.13273172). All open-source software is available on GitHub (github.com/baQFA).

## SUPPLEMENTAL MATERIAL

Supplemental material is available online only.
**TEXT S1**, DOCX file, 0.01 MB.
**FIG S1**, PDF file, 0.4 MB.
**FIG S2**, PDF file, 0.1 MB.
**FIG S3**, PDF file, 0.6 MB.
**FIG S4**, PDF file, 0.01 MB.
**FIG S5**, JPG file, 0.1 MB.
**FIG S6**, PDF file, 1 MB.
**TABLE S1**, DOCX file, 0.01 MB.

## ACKNOWLEDGMENTS

We thank Federica Andreoni and Markus Hilty for their support of the study and critical review of the manuscript and thank Mathilde Boumasmoud for providing the *E. faecium* strains and corresponding genomic information.

As this study did not include any human participants, it was exempt from ethics approval according to the Swiss Human Research Act.

All authors have given their consent to the publication of the manuscript.

None of the authors have declared conflicts of interest.

This work was supported by a Swiss National Science Foundation International Short Visit Grant (IZK0Z3_171414) to P.M.F., the National Institutes of Health through the National Institute of General Medical Sciences R01 GM117174 to K.P.L., the University of Zürich CRPP precision medicine for bacterial infections to A.S.Z., and Novartis Foundation for Medical-Biological Research fellowship 16B065, Swiss National Science Foundation and Swiss Foundation for Grants in Biology and Medicine P3SMP3_155315, and grant 1449/M from the Promedica Foundation to S.D.B., as well as a grant from the BéatriceEderer-Weber Foundation to S.D.B. and P.K.B.

Pascal M. Frey: principal investigator, acquisition of funding, planning of the study, data collection, statistical analyses, drafting of the manuscript. Julian Baer: planning of the study, data collection, statistical analyses, rewriting of the BaQFA R package, drafting of the manuscript. Judith Bergada-Pijuan: planning of the study, data collection, statistical analyses, BaColonyzer development, drafting of the manuscript. Conor Lawless: development of the original QFA method, drafting and intellectual review of the manuscript. Philipp Bühler: planning of the study, acquisition of funding, intellectual review of the manuscript. Roger D. Kouyos: planning of the growth modeling, intellectual review of the manuscript. Kathrine P. Lemon: planning of the study, acquisition of funding, intellectual review of the manuscript. Annelies S. Zinkernagel: planning of the study, acquisition of funding, intellectual review of the manuscript. Silvio D. Brugger: principal investigator, acquisition of funding, planning of the study, data collection, drafting of the manuscript.

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
