## [Reviewer comments · mSystems]

Quantifying variation in bacterial reproductive fitness: a high-throughput method

Pascal Frey, Julian Baer, Judith Bergadà Pijuan, Conor Lawless, Philipp Bühler, Roger Kouyos, Katherine Lemon, Annelies Zinkernagel, and Silvio Brugger

Corresponding Author(s): Pascal Frey, Bern University Hospital, University of Bern

Review Timeline:

Submission Date:	December 21, 2020
Editorial Decision:	December 26, 2020
Revision Received:	January 2, 2021
Accepted:	January 11, 2021

Editor: Matthew Traxler

Reviewer(s): Disclosure of reviewer identity is with reference to reviewer comments included in decision letter(s). The following individuals involved in review of your submission have agreed to reveal their identity: Matthew F. Traxler (Reviewer #3)

Transaction Report:

DOI: <https://doi.org/10.1128/mSystems.01323-20>

Revision of manuscript formerly entitled “Quantifying variation in bacterial reproductive fitness with and without antimicrobial resistance: a high-throughput method” (mSystems00500-20)

Response to Reviewers

We would like to thank the reviewers for their excellent and thorough work to improve our manuscript. Below, we provide responses to each of the reviewers' comments. All comments are re-stated in **bold** followed by our responses and description of the substance and location of any resulting changes made to the revised manuscript.

Reviewer #1

C1

I feel that the title is slightly misleading. Indeed, the authors find that in resistant strains reproductive fitness is higher. In only one of the two presented cases, isogenic strains are used and even in that case no mechanistic investigation regarding the drivers of fitness increase in the resistant strain is further pursued. Whilst, the authors use resistant strains to demonstrate the usefulness of their method, I think that it would be fairer to modify the title to more reflect the methodology and remove any mention of resistance. In this way, it is more likely that the study will be noticed broadly in the microbiology community, where it could also be useful in fields such as ecology.

We agree with the reviewer and have shortened the title accordingly to “Quantifying variation in bacterial reproductive fitness: a high-throughput method”.

Indeed, we had failed to mention the potential applicability of BaQFA to a broad spectrum of different fields in microbiology. We have thus amended the discussion section accordingly (lines 346 ff., page 16).

C2

Figure 3: it seems that replication has only been performed for 2 of the 3 strains of *S. aureus*. Is there a reason why the authors have not done this for all strains?

The development of our BaQFA adaptation focused on *S. aureus*, where we used the widely available lab strains JE2 and Cowan as the main subjects of study. We intend to use our BaQFA method to complement genomic data in the comparison of *S. aureus* clinical isolates in the investigation of clinical outcomes in patients with severe *S. aureus* infections. Therefore, we wanted to add one of our clinical isolates

to the comparison of measured normalised intensity values and actual counts of colony forming units (CFUs) from the culture spots. We initially thought that the comparison of the *S. aureus* clinical isolate would be better done in a more comprehensive manner including the genomic and clinical data, focusing on a different research question than we address in this manuscript. However, we agree with the reviewer that not providing fitness data on the clinical isolate may be considered problematic and have thus included fitness comparisons of the clinical isolate with Cowan and JE2, respectively (Supplement B Figure II).

C3

[Figure 3] It would be useful to add a second replicate for the *E. coli* DH5alpha, *E. faecalis* ST172 and *S. aureus* Cowan strains.

We fully agree and have added two more replicates of each strain to include three replicates of each presented strain (Figure 3).

C4

Figure 5: to strengthen any claims for increase in reproductive fitness in resistant strains, it would be important to add the metrics for the strains in isolation (similar to Figure 4), especially because the *E. faecium* backgrounds are practically isogenic.

We thank the reviewer for this suggestion and included the two *E. faecium* strains in a solo plate layout (Figure 5). However, it is important to note that the BaQFA method is primarily designed to compare strains growing on the same plate under the same conditions, and between-plate comparisons should be interpreted with caution due to random between-plate variability.

C5

Figures 4 and 5: there is quite some variability between replicates. In some cases, points from the same replicate seem to cluster (see R1 versus R2 and R3 in Figure 5). Could the authors comment on why they think that this is happening and how it could be taken into account when analysing the data? For example, if one looked at Tmax in Figure 5 they would reach a different conclusion based on replicate R1 and replicates R2 and R3. The same is the case for the other two metrics shown in this figure. It would be good if some caveats were discussed here and some suggestions were laid down for users of this method (for example how many replicates should one analyse ideally, and what errors they would expect). Whilst the correlation between the baQFA method and CFU counts shown in Figure 3 is quite convincing, the analysis of competitive fitness seems more prone to variability. As such, and since the authors mention the possibility of such competitive experiments as an advantage of the baQFA method (both in the abstract and in the discussion), I think it is important to further elaborate this point.

Indeed, we have observed substantial variability between experiments, which we suppose are due to small differences in starting conditions, with slight random variation in inoculum between plates being the most likely explanation. But also different plate conditions, like random differences of BHI concentration in agar and total BHI agar between plates may contribute to the between-experiment variation in

measured fitness. Those tiny differences at the start could be expected to have a potentially exponential impact on growth. However, the relative measures of two strains growing on the same plate alone or in competition is consistent even though absolute values might differ between different plates. That is the reason why we suggest to mainly focus on a comparative utilisation of BaQFA. In the future, a comparison of several strains/species always to the same comparator/reference strain might allow for a more generalisable approach.

We fully agree with the reviewer that this issue should be discussed in more detail, which we have now added to the revised manuscript (lines 414 ff., page 18).

C6

line 264: the figure citation seems to be wrong; "Figure 2" should be replaced with "Figure 3".

lines 274 and 281: the figure citations seem to be wrong; "Figure 3" should be replaced with "Figure 4".

line 286: the figure citation seems to be wrong; "Figure 4" should be replaced with "Figure 5".

The figure citation order has been updated accordingly now.

We apologize for these mistakes and feel deeply embarrassed they were missed after moving Figure 2 from the appendix into the main manuscript during our internal review of the manuscript.

Reviewer #2

Major comments

C7

(1) The authors insufficiently compare their method with existing methods. The authors claim that their methods are an improvement over existing methods for determining colony growth in bacteria. However, they do not perform a rigorous comparison with existing methods, except for a rather preliminary comparison between Colonyzer and BaColonyzer. Why are previously developed methods inadequate for estimate growth parameters in bacteria? How do methods of the authors compare with previous methods? If one would analyze the same dataset with different methods, would the method presented here provide more sensitive or accurate estimates of the growth parameters and why? Without a rigorous comparison, the reader is left uninformed about applicability of this work.

We are not entirely sure which previous methods the reviewer is referring to, in which fitness of bacteria may be quantified when growing in competition to each other. An existing and well established method of quantitative fitness analysis (QFA) in yeast has the drawback that its applicability to bacteria is limited, and has not been validated for bacterial growth [1,2]. Colonyzer is the software of this established QFA method, which we applied to our datasets to compare the results with our

BaColonyzer. To our knowledge, there is no other method or software that is suitable to be used for the type of raw data considered here (simple standardized time-lapse photographs of growth in a grid).

However, we absolutely agree with the reviewer that readers may desire a more general view of the results if other growth models or no model at all had been applied to our datasets. We have therefore included a re-analysis of our data with standard logistic and generalized logistic models, as well as a range of alternative fitness measures with and without using the Gompertz model (methods section lines 274 ff., page 13; results section lines 330 ff., page 15; and Supplement B Figure I and Figure III), and added a short discussion of the matter in our manuscript (lines 358 ff., page 16).

Furthermore, we have added information for a comparison of existing software to analyze microbial growth on agar plates to the introduction section to give readers a quick overview of how it compares to our BaQFA (lines 127 ff., page 6; and Table 1).

C8

(2) The authors use a simplified fitness measure. From what I understand from the methods, the authors only measure the mean pixel intensity inside the colony (after normalization) to determine the cell density. Previous methods estimate the colony biomass by accounting for both the pixel intensity and colony size. Why do the authors only account for mean pixel intensity and not for colony size? Would correlations in Fig. 3 improve when the authors would also account for the colony size?

Our new BaColonyzer algorithm does indeed already include culture spot size to derive intensity values. As this was not described well enough, we have now specified the processes of acquiring normalized intensity values, more clearly in our revised manuscript and included the description of additional features (lines 189 ff., page 10; and Supplement A).

C9

(3) The authors determine fitness by dividing the maximum growth rate by the time until maximum growth. The growth parameters are treated as being independent, but it has previously been shown that the maximum growth rate and lag time negatively correlate, making the implemented fitness proxy highly problematic. In addition, the authors do not determine if their fitness proxy has predictive value when directly competing strains in liquid culture. Finally, the authors do not discuss the low reproducibility between replicates for estimating the time until maximum growth (Fig. 4C, 5C) and maximum growth rate (Fig. 5B). High-throughput fitness estimates are only valuable to the extent that they predict the outcome of competition, the authors are therefore strongly encouraged to validate their fitness estimates.

As suggested by the reviewer, we have compared our BaQFA fitness measure with outcomes from liquid growth, which showed similar results (lines 275 ff., page 13; and lines 337 ff., page 15; Supplement B Figure III), even though these growth conditions are very different from BaQFA. However, we would also like to point out that our BaQFA method allows the calculation and derivation of a wide variety of measurements and parameters, and even different growth models can be chosen in the options of the BaQFA R package (a full list of parameters and description on

options is available within our open-source BaQFA R package on github.com/BaQFA).

We aimed to adapt a previously validated [1], versatile, inexpensive and easy-to-use method to measure fitness, and do not intend to redefine fitness. Thus, we have chosen the fitness measure outlined in our manuscript because it is simple and easy to use. We did not intend to treat maximum growth rate (MGR) and time to maximum growth (Tmax) as independent. We actually used both together because this way we increase the measurement scale, allowing us to better detect smaller differences in fitness that might have been missed when only looking at MGR or Tmax alone. Finally, we have now discussed between-plate variability in more detail (lines 413 ff., page 18), also see our answer to reviewer 1, comment C5.

Minor comments

C10

(1) "Reproductive fitness of bacteria is a major factor in the evolution and persistence of antimicrobial resistance and may play an important role as a pathogen factor in severe infections."

What do the authors mean by "reproductive fitness", how is this different from "fitness"?

We have used this term to clarify that our measure of fitness refers to reproduction alone, and does not include measuring any additional properties of the bacteria that may give them a superior (Darwinian) fitness in certain special niches or environments. We wanted a term that best described how fit bacterial strains or species are to reproduce in direct competition to each other.

C11

(2) "Reproductive fitness is the ultimate target of evolution, and in general, no cells can afford a reduction in fitness. Therefore, quantifying changes in fitness is highly informative about the evolutionary potential of cells. Antimicrobial-resistant strains of pathogenic".

Please choose your wordings carefully, both here and elsewhere in the manuscript, fitness is described as a phenotypic trait, which is incorrect. Fitness is a derived quantity that at best predicts the outcome of natural selection. In other words, fitness is a relative measure describing the success of one genotype over the other. It is therefore incorrect to refer to fitness as the "ultimate target of evolution".

We agree with the reviewer that fitness describes the success of one genotype (or strain) over the other. However, we somewhat disagree that it cannot represent a phenotype, as indeed what we measure with our BaQFA method is the phenotype of the strains or species as a proxy of success of its genotype. One important application of our method is to obtain a measure of the impact of bacterial genetic mutations on (phenotypical) growth, as it was done with the previously published QFA method for yeast [1], and which forms the basis for synthetic genetic arrays [3]. Nevertheless, we do understand the ambiguity of the term "ultimate target of evolution". We have thus rephrased that paragraph to remove it, and included the

reviewer's suggestion to state evolution as the success of a genotype in asexual organisms more clearly (lines 77 ff., page 5).

C12

(3) "The camera shutter was controlled through the LEGO Mindstorms robot and programmed to release every 10, 20 or 30 minutes after the robot had opened the plate lid (to reduce reflections and condensation artefacts; lid is needed to keep agar from drying)". It is unclear how the lid can be removed without the risk of contamination? It also remains unclear how the lid is handled by the robot. I urge the authors to provide complete information about their experimental setup to facilitate reproducibility. How does the LEGO Mindstorm setup look like, what parts are required and how is it programmed to control the Camera shutter?

The robot is set up in a closed PVC box inside the incubator. The purpose of the box is mainly to remove reflections from the metal interiors of most incubators, but probably further reduces any risk of contamination in case someone accidentally opens the incubator at the same time as the lid is opened. We have experienced almost no contaminations, which, according to the early time of first appearance on the images most likely happened before starting the QFA program. In case of relevant contaminations, which are easily identified on the agar plates, the experiment would need to be repeated.

As described in our manuscript (lines 170 ff., page 9), we used the LEGO Mindstorms EV3 set, thus all parts used for our robot are included in the EV3 set. To better illustrate the setup, we have added pictures of our first prototype in Supplement B Figure V. The camera shutter is released by the robot through mechanically short-circuiting a remote release cable (the normal way a Canon remote shutter works), of which the 2 ends are soldered to small copper plates, glued on two separate LEGO parts.

Our LEGO Mindstorms EV3 program for the robot is open source and freely available with all other BaQFA code (and 3D models for printing to make a BaQFA life easier) on our BaQFA GitHub page (github.com/BaQFA), which is described in the manuscript (lines 179 ff., page 9).

C13

(4) Fig. 2 - Graphics have unreadable axis.

We apologize for this; it is now corrected.

C14

(5) Supplementary Figure. Axes lack units. Horizontal dashed lines are not described?

We apologize for this; it is now corrected.

We hope to have been able to sufficiently answer the questions and issues raised by the reviewers, and are happy to provide any further information as needed.

Pascal M. Frey, MD, MSc
Staff physician
Bern University Hospital, Bern
Switzerland

Silvio D. Brugger, MD, PhD
Staff physician
University Hospital Zurich, Zurich
Switzerland

REFERENCES

[1] Addinall S, Holstein E-M, Lawless C, Yu M, Chapman K, Banks PA, et al. Quantitative Fitness Analysis Shows That NMD Proteins and Many Other Protein Complexes Suppress or Enhance Distinct Telomere Cap Defects. *PLoS Genetics* 2011;7:e1001362. <https://doi.org/10.1371/journal.pgen.1001362>.

[2] Lawless C, Wilkinson DJ, Young A, Addinall SG, Lydall DA. Colonyzer: automated quantification of micro-organism growth characteristics on solid agar. *BMC Bioinformatics* 2010;11:1–12. <https://doi.org/10.1186/1471-2105-11-287>.

[3] Tong A, Evangelista M, Parsons AB, Xu H, Bader GD, Pagé N, et al. Systematic Genetic Analysis with Ordered Arrays of Yeast Deletion Mutants. *Science* 2001;294:2364–8. <https://doi.org/10.1126/science.1065810>.

December 26, 2020

Dr. Pascal M Frey
Bern University Hospital, University of Bern
Department of General Internal Medicine
Inselspital
Bern
Switzerland

Re: mSystems01323-20 (Quantifying variation in bacterial reproductive fitness: a high-throughput method)

Dear Dr. Pascal M Frey:

I have reviewed your paper and am returning it to you for minor modifications. In principle, I am agreeing to accept your manuscript provided you address the points I have raised in my review. Many thanks for your re-submission to mSystems.

Below you will find the comments of the reviewers.

To submit your modified manuscript, log onto the eJP submission site at <https://msystems.msubmit.net/cgi-bin/main.plex>. If you cannot remember your password, click the "Can't remember your password?" link and follow the instructions on the screen. Go to Author Tasks and click the appropriate manuscript title to begin the resubmission process. The information that you entered when you first submitted the paper will be displayed. Please update the information as necessary. Provide (1) point-by-point responses to the issues raised by the reviewers as file type "Response to Reviewers," not in your cover letter, and (2) a PDF file that indicates the changes from the original submission (by highlighting or underlining the changes) as file type "Marked Up Manuscript - For Review Only."

Due to the SARS-CoV-2 pandemic, our typical 60 day deadline for revisions will not be applied. I hope that you will be able to submit a revised manuscript soon, but want to reassure you that the journal will be flexible in terms of timing, particularly if experimental revisions are needed. When you are ready to resubmit, please know that our staff and Editors are working remotely and handling submissions without delay. If you do not wish to modify the manuscript and prefer to submit it to another journal, please notify me of your decision immediately so that the manuscript may be formally withdrawn from consideration by mSystems.

Sincerely,

Matthew Traxler

Editor, mSystems

Journals Department
Reviewer comments:

Reviewer #3 (Comments for the Author):

1. Readers will come to the "Results" section before the "Materials and Methods". For this reason, please supply a section(s) in the "Results" that explain/reference Figures 1 and 2 in order.
2. P15 L315: Section 'E. faecium strains ST172 and ST172b': this section is currently extremely perfunctory. I would suggest additional text explaining the motivation for this experiment and an explanation of how related these two strains are at the genomic level. Additionally, text regarding what readers should take from the grid vs. solo experiment in Fig. 4 would be welcome.
3. For figures 3 and 4, please consider adding brackets above the violin plots with p values for the various pairs of measurements, so that readers can easily see where the data point to significant differences.
4. "Discussion". While the authors suggest that the BaQFA can be useful for a range of applications, and that this methodology comes with a series of caveats, it seems that it would be especially powerful when combined with mutational and/or genomic analysis. Such a powerful application seems worthy of comment here.
5. Similarly, the observations here, which point to the possibility that strains with antibiotic resistance may in fact possess higher overall fitness, run counter to the conventional hypothesis that such resistance would normally incur a cost leading to lower fitness. Such an observation merits interpretation here.
6. P20 L451: 'Availability of data'. Please move this section to the end of the Materials and Methods in order to comply with the ASM policy on data availability.

January 2, 2021

Revision of manuscript entitled “Quantifying variation in bacterial reproductive fitness: a high-throughput method” (mSystems01323-20)

Response to Reviewers

We would like to thank the reviewer for his excellent and thorough work to improve our manuscript. Below, we provide responses to each of the reviewer’s comments. All comments are re-stated in **bold** followed by our responses and description of the substance and location of any resulting changes made to the revised manuscript. All page and line numbers are referring to the marked-up manuscript.

Reviewer #3

1. Readers will come to the "Results" section before the "Materials and Methods". For this reason, please supply a section(s) in the "Results" that explain/reference Figures 1 and 2 in order.

We have now added an “Experimental Design and Workflow” section as an introduction/explanation referencing Figure 1 to the “Results” (P14 L289), and added an explanation and reference for Figure 2 in the following subsection “Derived normalised intensity values and colony-forming unit (CFU) counts” (P14 L305).

2. P15 L315: Section 'E. faecium strains ST172 and ST172b': this section is currently extremely perfunctory. I would suggest additional text explaining the motivation for this experiment and an explanation of how related these two strains are at the genomic level. Additionally, text regarding what readers should take from the grid vs. solo experiment in Fig. 4 would be welcome.

We fully agree and have added more information on the grid vs. solo experiments for the ST172/ST172b comparison (P15 L334) and in general (P9 L164). We also included a more detailed description of the differences of ST172 and ST172b and why we chose them to the revised manuscript (P15 L340).

3. For figures 3 and 4, please consider adding brackets above the violin plots with p values for the various pairs of measurements, so that readers can easily see where the data point to significant differences.

We have added brackets with p-values to the comparisons of competitive fitness as suggested. However, we would prefer not to add p-values for the individual components of fitness, as these are only meant to be supporting information for the final fitness measure. In addition, we have complemented the list of abbreviations in both respective figure legends to now include MGR and T_{\max} to make the distinction of these fitness components clearer (P30).

4. "Discussion". While the authors suggest that the BaQFA can be useful for a range of applications, and that this methodology comes with a series of caveats, it seems that it would be especially powerful when combined with mutational and/or genomic analysis. Such a powerful application seems worthy of comment here.

We absolutely agree with the reviewer and have included a short paragraph on using our BaQFA method with genomic analyses to the revised discussion section (P18 L408).

5. Similarly, the observations here, which point to the possibility that strains with antibiotic resistance may in fact possess higher overall fitness, run counter to the conventional hypothesis that such resistance would normally incur a cost leading to lower fitness. Such an observation merits interpretation here.

We thank the reviewer for this important comment. Indeed, acquisition of antibiotic resistance is generally linked to fitness cost and subsequent lower fitness under missing selection pressure (i.e., antibiotic exposure). Fitness costs might be alleviated by compensatory changes within the bacteria if they are maintained for generations [1,2]. In the case of VRE however, persistence of glycopeptide resistance is well described despite missing selection pressure [3,4]. Resistance containing plasmids can ensure their own maintenance in the absence of antibiotic resistance [3]. It has also been speculated that insertion sequence element insertions in the *vanA* gene cluster can result in fitness gain in the absence of glycopeptide exposure [2]. As the difference between the closely related ST172 and ST172b is not only the *vanA* cluster, but also several other predicted genes, it could be hypothesised that some of those other differences may provide an additional fitness gain or competitive advantage (e.g., bacteriocins). Thus, another very powerful application of our BaQFA method is its combination with genomic data analysis. BaQFA may be used to investigate changes of bacterial fitness of certain genomic variants, or the interaction of several genomic changes to determine compensatory mutations after antibiotic resistance acquisition. However, investigating these differences in more detail would have been beyond the scope of validating our BaQFA method.

We have added this information to the revised discussion section (P18 L396).

6. P20 L451: 'Availability of data'. Please move this section to the end of the Materials and Methods in order to comply with the ASM policy on data availability.

Done (P13 L283).

We hope to have been able to sufficiently resolve the issues raised, and are happy to provide any further information as needed.

Pascal M. Frey, MD, MSc
Staff physician
Bern University Hospital, Bern
Switzerland

Silvio D. Brugger, MD, PhD
Staff physician
University Hospital Zurich, Zurich
Switzerland

References

1. Starikova I, Al-Haroni M, Werner G, Roberts AP, Sørum V, Nielsen KM, Johnsen PJ. Fitness costs of various mobile genetic elements in *Enterococcus faecium* and *Enterococcus faecalis*. *J Antimicrob Chemoth.* 2013;68(12):2755–65.
2. Sivertsen A, Pedersen T, Larssen KW, Bergh K, Rønning TG, Radtke A, Hegstad K. A Silenced vanA Gene Cluster on a Transferable Plasmid Caused an Outbreak of Vancomycin-Variable Enterococci. *Antimicrob Agents Ch.* 2016;60(7):4119–27.
3. Hegstad K, Mikalsen T, Coque TM, Werner G, Sundsfjord A. Mobile genetic elements and their contribution to the emergence of antimicrobial resistant *Enterococcus faecalis* and *Enterococcus faecium*. *Clin Microbiol Infec.* 2010;16(6):541–54.
4. Johnsen PJ, Simonsen GS, Olsvik rjan, Midtvedt T, Sundsfjord A. Stability, Persistence, and Evolution of Plasmid-Encoded VanA Glycopeptide Resistance in Enterococci in the Absence of Antibiotic Selection In Vitro and in Gnotobiotic Mice. 2002;8(3):161–70.

January 11, 2021

Dr. Pascal M Frey
Bern University Hospital, University of Bern
Department of General Internal Medicine
Inselspital
Bern
Switzerland

Re: mSystems01323-20R1 (Quantifying variation in bacterial reproductive fitness: a high-throughput method)

Dear Dr. Pascal M Frey:

Your manuscript has been accepted, and I am forwarding it to the ASM Journals Department for publication. For your reference, ASM Journals' address is given below. Before it can be scheduled for publication, your manuscript will be checked by the mSystems senior production editor, Ellie Ghatineh, to make sure that all elements meet the technical requirements for publication. She will contact you if anything needs to be revised before copyediting and production can begin. Otherwise, you will be notified when your proofs are ready to be viewed.

Sincerely,

Matthew Traxler
Editor, mSystems

Journals Department
Supplement B: Accept

Supplement A - BaColonyzer Description: Accept

Supplement B: Accept

Supplement B: Accept

Supplement B: Accept

Supplement B: Accept

Supplement A: Accept

Supplement B: Accept